# Periodic Revisions of the International Choices Criteria: Process and Results

**DOI:** 10.3390/nu12092774

**Published:** 2020-09-11

**Authors:** Sylvie van den Assum, Rutger Schilpzand, Lauren Lissner, Rokiah Don, Krishnapillai Madhavan Nair, Ngozi Nnam, Ricardo Uauy, Yuexin Yang, Ayla Gulden Pekcan, Annet J. C. Roodenburg

**Affiliations:** 1Choices International Foundation, 2501 HE The Hague, The Netherlands; rutgerschilpzand@choicesprogramme.org; 2School of Public Health and Community Medicine, Institute of Medicine, University of Gothenburg, 405 30 Gothenburg, Sweden; lauren.lissner@gu.se; 3Nutrition and Dietetics Division, International Medical University, Kuala Lumpur 57000, Malaysia; RokiahDon@imu.edu.my; 4Formerly Department of Micronutrient Research, Indian Council of Medical Research (ICMR), National Institute of Nutrition, Hyderabad 500007, India; nairthayil@gmail.com; 5Department of Nutrition & Dietetics, University of Nigeria, Nsukka 410001, Nigeria; ngnnam@yahoo.com; 6Institute of Nutrition and Food Technology (INTA), University of Chile, 7810000 Santiago, Chile; druauy@gmail.com; 7National Institute of Nutrition and Health, Chinese Center for Disease Control and Prevention, Beijing 100050, China; yuexin_yang@sina.com; 8Department of Nutrition and Dietetics, Hasan Kalyoncu University, 27100 Gaziantep, Turkey; guldenpekcan@gmail.com; 9Department of Nutrition and Health HAS University of Applied Sciences, 5200 MA’s-Hertogenbosch, The Netherlands; A.Roodenburg@has.nl

**Keywords:** nutrient profiling, front-of-pack labeling, revision methodology, international choices criteria

## Abstract

Unhealthy diets contribute to an increased risk of non-communicable diseases, which are the leading causes of deaths worldwide. Nutrition policies such as front-of-pack labeling have been developed and implemented globally in different countries to stimulate healthier diets. The Choices Programme, including the International Choices criteria, is an established tool to support the implementation of such policies. The Choices criteria were developed to define the healthier choices per product group, taking saturated fatty acids, trans fatty acids, sodium, sugars, energy, and fiber into account. To keep these criteria updated, they are periodically revised by an independent international scientific committee. This paper explains the most important changes resulting from revisions between 2010 and 2016 and describes the process of the latest revision, resulting in the International Choices criteria version 2019. Revisions were based on national and international nutrition and dietary recommendations, large food composition databases, and stakeholders’ feedback. Other nutrient profiling systems served as benchmarks. The product group classification was adapted and new criteria were determined in order to enhance global applicability and form a credible, intuitively logical system for users. These newly developed criteria will serve as an international standard for healthier products and provide a guiding framework for food and nutrition policies.

## 1. Introduction

Non-communicable diseases (NCDs) such as cardiovascular diseases, diabetes, and cancers are the causes of over 70% of deaths worldwide. NCDs are a major public health problem, especially in low- or middle-income countries, as over 75% of the worldwide NCD deaths occur in these countries. NCDs share behavioral risk factors that are, for a large part, modifiable, such as sedentary lifestyles and unhealthy diets [1]. Limiting total energy intake and levels of sugar, saturated fatty acids (SAFAs), trans fatty acids (TFAs), and sodium while consuming adequate amounts of essential nutrients, such as vitamins, minerals, and fiber contributes to a healthy diet. However, for consumers, it is often impossible to know which products fit best in a healthy diet without comparing the detailed nutrition labels on the back of the package [2].

Many nutrient profiling systems are currently in place and are the basis for, among other purposes, front-of-pack nutrition labeling (FoPNL) [3]. The Choices Programme is a nutrient profiling system based on the International Choices criteria (“Choices criteria” hereafter) which, per product group, indicates the healthier food options with regards to NCD prevention by setting criteria for SAFAs, TFAs, (added and total) sugar, sodium, fiber, and energy. The Choices criteria only address the levels of these six nutrients. All other health aspects of food products, such as food safety and the presence of additives, artificial sweeteners, and potential allergens are to be regulated by national food legislation. The system helps health authorities in shaping food and nutrition policies, and it facilitates consumers to change their diets by making healthier choices easier. Moreover, the Choices criteria encourage food and beverage producers to enhance the nutrition quality of their products, resulting in positive product changes by which consumers benefit without changing their dietary patterns [4,5]. Nutrient profiling systems similar to the Choices Programme, i.e., systems that indicate the healthier choice per product group, are implemented widely and include the Nordic Keyhole scheme and the Healthier Choice Symbol of Singapore, Thailand, Malaysia, and Brunei, as well as and several National Heart Foundation franchises (Finland, Slovenia, Nigeria, and South Africa) [6,7,8,9].

The development of the Choices criteria was previously described by Roodenburg et al. in 2011 [10]. In short, the development of the criteria was a response to the WHO Global Strategy on Diet, Physical Activity, and Health (2004) [11]. Consequently, the criteria setting was based on WHO dietary recommendations for the prevention of nutrition-related chronic diseases and was designed to select the top 20% most healthy products within a product group for basic foods and the top 10% for non-basic foods. Basic product groups contribute to the intake of essential nutrients, e.g., fruits, vegetables, dairy products, and bread. Non-basic product groups are discretionary and include snacks and sauces. Snacks are defined as products consumed in between meals and that do not belong to any other product groups. To determine these percentages, an international food composition database was used [10]. All products, except for those containing >0.5% alcohol, food supplements, products for use under medical supervision, and foods for children under one year old can be categorized within one of the described product groups. For each product group, specific nutrient cut-off values were developed. Notably, these 10% and 20% figures served as a guide rather than strictly set values. Decisions on the definition of the product groups and criteria were based on (a) the availability of healthier alternatives within product groups, (b) the opportunity to stimulate product innovation, and (c) alignment with dietary recommendations [10]. For some product groups, like fresh fruit and vegetables, all products comply by default because of their well-established beneficial effects [12].

Periodic revision is a fundamental characteristic of the Choices criteria and an important tool to stimulate a process of stepwise product improvement by reduction of SAFAs, TFAs, sodium, and sugar. Since the publication by Roodenburg et al. in 2011, several major revisions of the Choices criteria have taken place. To allow time for the industry to reformulate their products, the criteria revision takes place once every four years (Figure 1). The revision also creates the possibility to adapt the criteria based on practical experiences, recent scientific research, and progress in product reformulation. The Choices International Scientific Committee (ISC) independently oversees all revision processes, and it approved several main changes up to the version of 2016. This paper first describes the main decisions resulting from the revisions until 2016. A detailed overview of all changes up to 2016 can be found in Appendix A. The second part of this paper explains in detail the methodology and outcomes of the most recent criteria revision of 2018, which resulted in the 2019 version of the Choices criteria.

## 2. Modifications of the Choices Criteria between 2010 and 2016

### 2.1. Added Sugar and Total Sugar

The criteria for sugar were initially expressed as added sugar. Added sugars are defined as all monosaccharides and disaccharides with a caloric value of >3.5 kcal/g that are derived from sources other than fresh fruits and vegetables and milk products. Naturally occurring sugars were left out of consideration because these are often impossible to remove from products, leaving limited opportunities for reformulation. In many countries, nutrition declarations only display total sugar values, and available sugar level data are often derived from these labels. Given the insufficient data on added sugar levels, cut-off values for total sugars were also calculated to use for product assessment when data on added sugar are not available. For each product group, mean naturally occurring sugar levels were calculated and added to the added sugar criteria to obtain the total sugar level. For example, the natural sugar content for ‘Breakfast cereal products’ is, on average, 2 g/100 g, calculated using data from the USDA Food Composition Database [13]. The criterion for added sugar was calculated to be 17.5 g/100 g, resulting in a criterion for total sugar for ‘Breakfast cereal products’ of 19.5 g/100 g.

### 2.2. Units of Expression: Grams per 100 g

Cut-off values were originally expressed in (m)g/100 g, (m)g/100 kcal, en% (nutrient energy/total energy in a product *100%), or % of total fat (in case of the SAFA criteria). To standardize the criteria for coherence with the units of expression used on the back-of-pack nutrition panel and to make them comparable with other nutrient profiling systems, all cut-off values were transformed into (m)g/100 g [14,15]—except for kcal/portion, which is only used for energy cut-off values in meals, sandwiches/rolls, and snacks, as the products in these categories are typically consumed per portion.

Each method of expression has its strengths and weaknesses. The expression of the criteria in (m)g/100 g prevents the addition of calories to products by the industry in order to meet the criteria, which was an unfavorable side effect when maximum criteria were expressed in (m)g/100 kcal.

By using g/100 g rather than g/portion, the Choices criteria put emphasis on the importance of product quality over quantity, as the expression in g/100 g encourages the industry to enhance product composition and not to simply reduce the specified portion size in order to meet the criteria. There are currently no global standards for portion sizes specified for product groups. This makes it possible to claim smaller portion sizes than those commonly consumed and thereby comply with the criteria. The use of distinct product groups in the Choices criteria also reduces the need to focus on portion sizes within specific product groups, e.g., breakfast cereals, as portion sizes of products within the same product group can be approximately the same.

To convert the former expression units (en%) into grams per 100 g, detailed product data from the Dutch Choices Product Database and USDA Food Composition Database were used and plotted into a graph. Appendix A describes an example for the product group ‘Main courses.’ Different main courses were plotted by their SAFA content expressed in en% on the y-axis and in g/100 g on the x-axis. A linear relationship was determined, and the former cut-off value for SAFA, which was 13 en%, was converted into 2 g/100 g.

### 2.3. Changes in Product Group Classifications

Modifications were made to the product group classifications. The product group ‘Nuts and seeds’ was newly added to the basic food category as a specific food group because of their important role in diets in many countries [16]. Though nuts and seeds have a high caloric content, they confer beneficial effects on human health, and consumption is promoted in an increasing number of national dietary guidelines [17,18,19,20].

Composite dishes initially consisted of the product groups ‘Main course’ and ‘Filled sandwiches and rolls.’ To select healthier options in catering and the ready-to-eat market, the product groups ‘Small meals’ and ‘Salads’ were added as separate groups.

‘Fruit juices’ were moved from basic product groups to non-basic product groups because of their high sugar content. The consumption of sugar-sweetened drinks has been observed to be associated with weight gain [21,22]. Moreover, liquids have shown a weaker satiety response than solid foods [23], which may result in an enhanced overall calorie consumption from sugar-sweetened and other caloric beverages. Therefore, the ISC decided to have only plain water, tea, coffee, and milk (products) in the basic foods’ category.

A distinction within the group of grains and cereal products, between wheat and non-wheat products, was decided when the total sugar cut-off values were added. The average amount of naturally occurring sugar in non-wheat products appeared to be considerably higher than in wheat products (8.5 versus 2 g/100 g). Two different levels of total sugar were needed, and, therefore, the distinction between wheat and non-wheat products was made.

Lastly, the product group ‘Processed beans and legumes’ was added. Previously, these products were part of the ‘Processed fruit and vegetables’ product group. However, the sodium level of 100 mg/100 g resulted in the non-compliance of all processed beans and legumes. As the consumption of beans and legumes are to be encouraged [2], a new product group was created with a sodium cut-off value of 200 mg/100 g.

## 3. Revision 2018—Methods

The major aim of the 2018 revision was to evaluate product group classification, to determine new (often stricter) cut-off values when appropriate, to enhance global applicability, and to form an intuitively logical system for consumers. The Choices criteria can be used by existing franchise initiatives to align their criteria and thereby enhance coherence and facilitate international trade. The newly developed criteria provide an up-to-date benchmark to evaluate the healthiness of products and product ranges to support governments and the industry in their food, nutrition, and health strategies.

This work was conducted between September 2017 and January 2019, and it was directed by the ISC, which consists of leading independent scientists from Africa, Asia, South America, and Europe [24]. The independence of the committee is guaranteed because their decisions are made autonomously without the interference of the industry or any other interest groups or from the board of the Choices International Foundation. The committee met twice in person during this period and had continuous discussions by teleconferences and written consultations. The Choices global secretariat was responsible for the preparatory work to support the decisions of the ISC. The Choices Scientific Advisory Group, which consists of nutrition experts from food companies that are globally associated with Choices International, indirectly advised on the practical feasibility of the criteria from the industry perspective by providing the Choices secretariat with their feedback upon request.

The revision followed the steps displayed in the workflow chart shown in Figure 2.

### 3.1. Stakeholder Evaluation

At the start of the criteria revision, a criteria evaluation form was sent to internal and external stakeholders. Responses were received from members of the Choices ISC, national scientific committees (Belgium and Czech Republic), the Brunei Scientific Committee (Brunei), nutrition experts of The George Institute (Australia), and the industry (Arla Foods, Danone, and Unilever). The feedback was reviewed during the first ISC meeting of this periodic revision and served as input for further evaluation. In these responses, suggestions were made on product classifications, identifying products as basic or non-basic, units of expression, cut-off values, and the inclusion of additional nutrients to assess.

### 3.2. Review of Changes in Dietary Recommendations

To be aligned with the latest views on healthy diets, an inventory was made of organizations that provide dietary guidelines and/or nutrition recommendations. New versions published since 2013 were identified and compared with former versions. New recommendations on macronutrient values were published by EFSA (Scientific Opinion on Dietary Reference Values for Energy, 2013 and Scientific and Technical Assistance on Trans fatty Acids, 2018), WHO (Sugar intake for adults and children, 2015 and REPLACE trans fat, 2018), USDA (Dietary Guidelines for Americans, 2015–2020), The American Heart Association (Dietary and Lifestyle Recommendations, 2016), Chinese Nutrition Society (The Chinese Dietary Guidelines, 2016), and Public Health England (Eatwell Guide, 2016). Macronutrient recommendations from the investigated organizations on total fat, saturated fat, sodium, and fiber had not changed since 2013. However, USDA and Public Health England revised their sugar recommendations from “Reduce the intake of calories from added sugars” and “Cut down on sugar” to, respectively, “<10% of calories per day” (aligning with WHO recommendations) and “<30 g per day.” An energy intake varying between 1800–2600 kcal is recommended, dependent on energy expenditure, gender, and age. Previously consulted guidelines (EFSA, Public Health England, and Chinese Nutrition Society) had not specified energy levels. Appendix A gives an overview of the changes. The recommendation on the maximum intake level of TFAs became stricter. These changes in recommendations were incorporated into the ISC discussions. For example, lowering of TFA cut-off values was considered for several product groups to align with the stricter guidelines for TFA.

### 3.3. Review of Product Group Definitions

The aim of the present revision of the Choices criteria was to make the criteria more internationally applicable by taking the different global food cultures into consideration. Consequently, new product groups from Asian and African food cultures were introduced and several product group definitions were adjusted. The review and conclusions on the definition of product groups were based on expert opinions of the ISC members and external stakeholders. To ensure a credible system, the number of product groups had to be limited and decisions about the creation of new groups, the combination of existing groups, and the revision of product group definitions had to be dependent on the decision framework described by Roodenburg et al. [10]. The most important considerations are the consumer understanding of the system, the optimal stimulation of reformulation by the industry, and global applicability. This was assessed by the expert opinions of the members of the ISC.

### 3.4. Databases

The Choices criteria aim at identifying the 20% healthiest products within the basic product groups and the 10% healthiest products in the non-basic product groups. The exceptions to this rule are the 100% compliance of fresh fruits and vegetables and a large majority in the product groups of ‘water’ and ‘fresh seafood’. To identify this 20% or 10% level, a comprehensive overview of the product composition in the different markets was needed as a reference. Two recent food product composition databases, containing the nutrient compositions of 22,697 and 45,593 different products that are currently on the market, were used to analyze the strictness of the criteria. The databases were obtained from The George Institute for Global Health, an Australian independent medical research institute [25]. The databases contain branded data from 8 most important markets (Australia, China, Hong Kong, India, New Zealand, South Africa, UK, and USA) in different continents to obtain global coverage. For practical reasons, only databases in the English language were used. The two databases—named GI1 and GI2 in order of date obtained—differ in size, composition, and origin (Table 1 and Appendix A). GI1 contains data from the 22 largest global food and beverage producers. The nutrient data were collected with a smartphone application that captures information by taking photos of the front and back panel of packaged food products in grocery stores and other retailers in the countries mentioned above. These data were then stored in the George Institute’s FoodSwitch database. This database was checked and supplemented with data by these 22 companies. For some product groups, the amount of available data was low. Therefore, additional data were obtained (GI2 database), including data from the FoodSwitch database with data from local and global food producers. The data of both databases were grouped into Choices product groups and checked for consistency by The George Institute.

USDA National Nutrient Database for Standard Reference (2018), Indian Food Composition Tables (2017), and the Dutch Food Composition Database (NeVo 2016) were used as references and an additional source of product data in instances when the GI databases were inconclusive and insufficient to address the products under discussion [13,26,27]. Compliance percentages were calculated with data from these databases and compared to the outcomes from the GI1 and GI2 databases.

### 3.5. Calculation of Products’ Compliance to Current Cut-Off Values

Predetermined ‘critical nutrients’ were used to calculate product compliance (percentage of products in the GI databases that comply with the product group’s critical nutrients criteria). For this, critical nutrients were specifically defined by expert opinions of the ISC members for each product group, ultimately forming the most relevant and usually the most limiting factor for compliance. Table 2 displays the critical nutrients per product group.

The percentage of products that complied to the criteria was calculated by compiling lists of all products from all countries for each product group. Products were excluded if data on one or more of the critical nutrients were lacking, except when nutrient data were lacking for the complete dataset (fiber data in GI2 and TFA in both databases). Graphs were used to visualize the distribution of the products’ nutrient levels and to demonstrate the effects of changing cut-off values on the compliance level. The latter was needed to facilitate the discussion on the cut-off values in the meetings of the ISC. Appendix A illustrates an example of such a graph. The compliance to the current criteria was separately calculated in both GI1 and GI2 databases (taking all critical nutrients into account). When data on certain critical nutrients were not available for the full dataset (fiber data in GI2 and TFAs in both databases), the compliance was calculated without that nutrient, with the acknowledgement that the compliance might be lower than calculated.

### 3.6. Review of Product Group Nutrient Cut-Off Values

If the percentage of compliance described in the previous paragraph was well above 20% for basic food groups or above 10% for non-basic food groups, new cut-off values were calculated that neared these percentages. Again, when fiber or TFAs were not available for the full dataset, the compliance was calculated with the available critical nutrients. A lower actual compliance was assumed and taken into consideration when setting cut-off values. In some instances, higher cut-off values were considered in order to identify more products as healthier choices (compliant to the franchise criteria) because these products are recommended for consumption by leading health or nutrition organizations, such as WHO or US FDA. For example, the maximum sodium level of fresh seafood was raised to promote the consumption of products from this product group. In some cases, the results from the two GI databases varied in compliance. Causes for these differences could be the origin of the database (data from products of large food companies versus data from products of all different companies), the number of products in the database for the particular product group, and the fact that fiber data were lacking for GI2. In these cases, the USDA database was referred to. The compliance of the USDA data was calculated and compared with the outcomes from the GI1 and GI2 databases in order to form an informed conclusion on the new cut-off value.

When considering making the criteria more stringent, positive indicator products and products that are iconic for a healthy diet were analyzed against the proposed criteria to check for compliance. For example, oats and oatmeal are products that contribute to a healthy diet. The SAFA level of unprocessed oats is 1.2 g/100 g, which does not meet the SAFA criteria for the grains group. The SAFA level of this group was raised to 1.2 g/100 g to make oat comply. The Scientific Advisory Group was consulted by the secretariat to obtain feedback on the feasibility of the proposed new criteria.

Sugar data in the GI databases only consisted of total sugar data and did not specify added sugar. When new criteria were set for total sugar, added sugar changed accordingly. The difference in total and added sugar from the 2016 criteria were used to calculate the new criteria for added sugar. These differences were previously calculated by methods described above. For example, for breakfast cereal, the total sugar level was reduced from 19.5 to 17 g/100 g. The added sugar cut-off value was accordingly lowered from 17.5 to 15 g/100 g.

### 3.7. Comparison of Cut-Off Values of Other Positive Labeling Systems

To contribute to international convergence of the different criteria sets of front-of-pack organization schemes and to learn from other nutrient profiling systems, the International Choices criteria were, per product group, compared with criteria of the other labeling schemes that use product group specific criteria. These included the Belgian Choices criteria, the Dutch Choices criteria, the Czech Choices criteria, The Finnish Heart Foundation, the Nordic Keyhole scheme, the Nigerian Heart Foundation, Zambian Good Food, Chinese Nutrition Society, Healthier Choice Malaysia, Healthier Choice Singapore, and Healthier Choice Thailand. For each product group, it was checked whether product classification was comparable, and, in such case, the cut-off values of these positive labeling systems were used as a benchmark. For example, when a cut-off value change was considered for a certain product group, it was checked whether the new value deviated substantially from cut-off values in other nutrient profiling systems. All the proposed new cut-off values fell within an acceptable range compared to the different other criteria sets.

## 4. Revision 2018—Results

### 4.1. Product Group Classification

To improve global applicability and encourage product reformulation, the classification of products was changed for a number of product groups. An overview of the changes is shown in Table 3.

#### 4.1.1. Sources of Carbohydrates

The category ‘sources of carbohydrates,’ consisting of multiple product groups, underwent a major classification change. Through this change, products that are regularly consumed in Asia or Africa were included in order to further encourage the reformulation of these products. Firstly, the product group ‘Unprocessed potatoes’ was broadened to ‘Unprocessed tubers used as staple food’ to include regularly consumed products such as cassava, yam, and taro [28]. Secondly, the ‘Potatoes (processed), pasta, and noodles’ group was subdivided into ‘Processed tubers used as staple,’ ‘Plain noodles and pasta,’ and ‘Flavored noodles and pasta.’ Flavored or instant noodles are a major product group consumed in Asia and increasingly so in the rest of the world, and they are an important contribution to total sodium intake [29,30]. By creating a separate product group for these products, the industry will be stimulated to lower the sodium content of flavored noodles, guiding consumers to lower sodium options. Thirdly, the product groups ‘Rice,’ ‘Grains and cereal products (wheat-based),’ and ‘Grains and cereal products (non-wheat)’ were combined into one product group labeled ‘Grains.’ The latter two product groups had similar cut-off values and could, therefore, be combined. Within a diet, rice is often interchangeable with other types of grains, such as barley, quinoa, and freekeh, and it was therefore decided to include rice into the ‘Grains’ product group. This added to the objective to create an intuitively logical system for consumers.

#### 4.1.2. Meat and Meat Substitutes

Our aim to make the criteria more globally applicable resulted in the removal of meat substitutes from the ‘Processed meat’ product group. Meat substitutes are difficult to objectively define and were defined by intended use and consumer perception. These can differ largely from one food culture to another. Many products that are perceived as meat substitutes in some food cultures are not consumed as such by others. For example, in Western food cultures, tempeh and tofu products can be perceived as meat substitutes, while in many Asian countries, they are not [31]. It was therefore decided to reclassify meat substitutes by their main ingredient; thus, dairy-based meat substitutes were classified as ‘Milk (products) and legumes-based products as ‘Processed beans and legumes.’ This also solved an issue of these products overlapping with two product groups, e.g., tofu belonging to both ‘Meat substitutes’ and ‘Processed beans and legumes.’

#### 4.1.3. Seafood

Seafood was previously divided into two product groups: ‘Fresh or fresh frozen fish, shellfish, and crustaceans’ and ‘Processed fish or fish products.’ After elevating and lowering the sodium cut-off value for these two product groups, respectively, the criteria became the same. This resulted in a merge of these product groups into ‘Fresh, frozen, or processed seafood.’

#### 4.1.4. Edible Insects

Edible insects were previously not defined as a Choices product group, even though insects have been a common part of the diet in some cultures and edible insect consumption has been growing globally [32]. A separate product group was therefore created for edible insects, following the example of the Choices criteria of Zambia [33].

#### 4.1.5. Milk and Milk Substitutes

The former ‘Milk (products) and substitutes’ was divided into ‘Milk (products)’ and ‘Non-dairy milk substitutes.’ Several non-dairy milk substitutes, in particular those not fortified, contain a limited diversity of nutrients compared to dairy milk products while containing less sugar [34]. As a result, non-dairy milk substitutes comply as much as or even better than milk products while not contributing as much to the micronutrient intake of consumers. Therefore, a separate product group was created for ‘Non-dairy milk substitutes’ with stricter cut-off values for SAFAs and sugar. It was decided to add this product group as a non-basic group, since most milk substitutes do not comply with the equivalence criteria for ‘Milk (products).’ Equivalence criteria describe the minimum level of micronutrients that a basic product group should contain (Appendix A).

#### 4.1.6. Meals

It appeared to be difficult, in practice, to clearly distinguish between ‘Main courses’ and ‘Small meals,’ and the energy criteria for these two product groups showed an overlap. Therefore, these two groups were combined into a single new product group called ‘Main meals,’ now including all meals consumed for breakfast, lunch, and dinner. The meal category was further simplified by removing ‘Mixed salads,’ which was defined as ‘Salads containing at least 70% fruits or vegetables.’ This means that ‘Mixed salads’ can be classified as ‘Processed fruits and vegetables.’ As a general rule, products belong to a certain product group when 70% of that product matches the definition of the product group [35]. Salads that consist less than 70% fruits or vegetables were now classified under ‘Main meals.’ Furthermore, for many cultures, soup is a main constituent of the meal and an important source of nutrients. The product group ‘Soups’ was, therefore, moved from the non-basic product groups to basic product groups as a subgroup within the ‘Meals’ category.

#### 4.1.7. Water-Based Sauces and Snacks

The rationale behind dividing ‘Other sauces (water-based)’ into ‘Dark sauces’ and ‘Other sauces (water-based)’ and ‘Snacks’ into ‘Savory snacks’ and ‘Sweet snacks’ was to guide consumers in finding healthier options in these new product groups and to encourage product reformulation by the industry. For example, black sauces like soy sauces usually contain very high levels of sodium that significantly contribute to the total sodium intake in certain countries [29]. Criteria based on the top 10% healthiest products for all water-based sauces will include a lower sodium cut-off value than is attainable for many dark sauces. Identifying the top 10% heathiest dark sauces will encourage the producers of dark sauces to reformulate to make healthier choices available for consumers.

#### 4.1.8. Bread Toppings

The decision to delete the ‘Bread toppings’ product group was also based on enhancing the global applicability of the criteria. The definition of this group is also based on intended use rather than on product composition and consequently dependent on food cultures. In addition, the variation of products within the product group was large, making them difficult to compare and to set suitable criteria. Therefore, the different bread toppings were incorporated into the groups of their main component. Jams are now classified as processed fruits and vegetables, while peanut butter now belongs to the group of processed nuts and seeds.

Despite the reclassifications, the total number of product groups remained unchanged (32) after the revision (Table 4).

### 4.2. Cut-Off Values

The maximum values for sugar, sodium, and saturated fat were evaluated as described above. This resulted in the lowering of a number of cut-off values. Sugar maximum values were lowered in four product groups (‘Breakfast cereal products,’ ‘Soups,’ ‘Emulsified sauces,’ and ‘Beverages’). Sodium was lowered in seven product groups (‘Processed meat and meat products,’ ‘Processed seafood,’ ‘Cheese,’ ‘Soups,’ Meal sauces,’ ‘Emulsified sauces,’ and ‘Sweet snacks’) The SAFA cut-off values were lowered in three product groups (‘Cheese,’ ‘Emulsified sauces,’ and ‘Savory snacks’).

More lenient cut-off values were set for SAFAs for the ‘Grains’ product group. This decision was made to cover all sorts of grains including oats. The sodium cut-off value for fresh fish was elevated from 130 to 300 mg/100 g to make all kinds of seafoods compliant to the criteria, including crustaceans and mollusks. By elevating the sodium cut-off value for fresh fish and lowering the sodium for processed fish, the criteria for both product groups became equal. Consequently, to enhance simplicity, the two product groups were merged into one group.

This criteria revision carefully reconsidered the current TFA cut-off values, as in recent years, where global authorities have been emphasizing their concern on TFA intake. The WHO is urging governments to urge the elimination of industrially produced TFAs in food products. This elimination should be completed by 2023 [36]. The US FDA announced that they no longer recognize industrial TFAs as GRAS (Generally Recognized as Safe), meaning that TFA is no longer considered a safe food component [37]. Considering the FDA definition for ‘zero TFA’ for label claims to be 0.5 g/100 g and the Chinese standard for zero TFAs is 0.3 g/100 g, the majority of the Choices cut-off values for TFAs were already quite conservative. Product groups that had cut-off values of higher than 0.1 g/100 g were considered for TFA reduction to meet the standards set by these health authorities, using the Chinese criteria as an intermediate step. Therefore, ‘Oils, fats, and fat-containing spreads’ decreased from 1.0 g to 0.5 g/100 g, and ‘Emulsified sauces’ decreased from 0.35 to 0.3 g/100 g. TFAs in the snacks category was maintained at 0.4 g/100 g after concluding that, with the current state of innovation, it is not yet possible to reduce to 0.3 g/100 g of TFAs for many snacks in the market, in particular because of the preference of oils with a high level of poly-unsaturated fatty acids. The Scientific Committee views these values as intermediate steps towards the further elimination of TFAs in the next criteria revision (2022).

Cut-off values were set on approximately 20% for basic product groups and 10% for non-basic product groups. There are some product groups that deviate from this value. For ‘Fresh or fresh frozen fruits, vegetables, and legumes’ and ‘Plain tubers used as staple,’ there is a 100% compliance because all products comply by default. Almost all ‘Plain water, tea, and coffee’ products comply (97%). For the product group ‘Fresh, frozen, or processed seafood,’ the compliance is close to 40%.

After consultation with several actors within the insect processing and retail sector, it appeared there were insufficient data available to be able to calculate reliable compliance percentages. The Zambian criteria for ‘Unprocessed insects’ were therefore used as a benchmark to set the international criteria. A small nutrient database containing various insects was formed by collecting and combining data from various publications [38,39]. This led to the decision to elevate the sodium level from 100 to 200 mg/100 g so that crickets could comply with the criteria. The revised criteria are shown in Table 4.

## 5. Discussion

This article summarizes revisions between 2010 and 2016, and it reviews the detailed process and outcome of a typical periodic criteria revision process for the Choices Programme that is conducted every four years. This revision took place in 2018.

Various different front-of-pack and nutrient profiling systems have been developed in the last decade. Comparisons between systems are discussed elsewhere [3,40,41]. The Choices Programme is characterized as a voluntary system with a high level of simplicity to be useful for quick decisions and to be interpretable for everyone, including people with low literacy skills. It has a positive approach, meaning that the program encourages the consumption of healthier foods as opposed to discouraging unhealthy products. By being positive and simple, its implementation supports the WHO slogan “Make the healthy choice the easy choice” [42]. The program has several applications. It can (a) help consumers identify healthier options within the product group by displaying logos on packaged foods, (b) set goals for the industry to enhance nutrition quality of food products, (c) support companies to evaluate the healthiness of their product range, and (d) serve as a tool for governments to implement their nutrition policies. This multi-purpose character introduces challenges when defining product groups and setting criteria. Because the maximum levels of sugar, sodium, SAFAs, and TFAs are determined by the rule that 10% or 20% of the products in the category should comply, differences in maximum values may occur that seem counterintuitive. For example, criteria for snacks are more lenient than criteria for milk products. A sweetened milk product would not be eligible for certification, while ice cream (considered a non-basic food) with a reduced sugar level possibly would. The criteria for snacks and other non-basic food groups can therefore be a challenge to communicate to consumers as a heathier choice but are, on the other hand, necessary to encourage reformulation and to indicate a ‘healthier’ or ‘less unhealthy’ option in the specific category. This challenge of consumer communication could be addressed by incorporating the name of the product group into the logo. In this way, it will be made clear to consumers that the logo-bearing product is healthy relative to other products in a specific product group. The Thai Healthier Choice organization adopted this strategy [43]. Another way would be to exclude all non-basic product groups, as applied within the Keyhole scheme [44]. However, this eliminates the opportunity for the reformulation of a large number of less healthy foods.

The Choices International criteria are evaluated every four years to keep in pace with scientific insights, consumer taste, and developments in product reformulation. Based on calculations from the large-scale databases on the nutrition composition of individual food products and expert opinions from members of the Choices International Scientific Committee, several maximum levels of nutrients to limit were lowered. This stepwise lowering of the criteria by each revision can lead to the enhanced encouragement of the food industry to improve their products in order to comply with the new criteria [45].

The setting of an updated maximum value for sugar in the ‘Beverages’ product group brought about a challenging discussion. Due to the large quantity of products within this product group that contain non-nutritive sweeteners to enhance taste and that do not contain any sugar, the compliance percentage of 10% envisioned for non-basic product groups resulted in a sugar level of zero. Instead of following these calculations and setting the maximum level for sugar at zero, the International Scientific Committee decided to lower the maximum sugar level from 5 g of total sugar per 100 g (translating to 20 kcal/100 g) to 2.5 g (translating to 10 kcal/100 g). This was decided for two reasons. The first was to give a signal to the beverage industry that further sugar reduction in beverages is needed. The second was to show consumers an alternative for beverages high in sugar other than options containing non-nutritive sweeteners. The 2.5 g/100 g value corresponds with the EU regulation on claims for low sugar [46].

Whether or not to prevent beverages that contain non-nutritive sweeteners (NNSs) to be able to comply was discussed in-depth during this revision. There is much debate on the risks and benefits of the use of NNSs. FAO/WHO Joint Expert Committee on Food Additives (JECFA) and other international authorities have established an acceptable daily intake (ADI) for most artificial sweeteners that are well above the daily intakes of consumers [47]. The ISC observed that the data on adverse effects of NNSs, such as possible effects on the gut flora or on appetite or on the development of a sweet taste, are non-conclusive [48,49,50]. This led to the decision to currently keep NNS-containing drinks eligible to the criteria while still monitoring the research developments in this area. The scientific body of evidence on this item has been on the agenda of the previous revisions of the Choices criteria and will be re-evaluated in the next criteria revision.

### 5.1. Limitations

The Choices criteria are set according to a 10–20% compliance in the market. When revising the global criteria, a global product composition database containing a representative selection of products from all countries would ideally be used. In the absence of such a database, a composite database has been developed from eight countries well-distributed over the globe, for which a database in English was available. Not all countries were evenly represented in this database. For example, data from the USA were much more abundant than Indian data. Separately calculating compliance for each country and then combining data afterwards was considered. However, this would result in giving too much value to a small number of products. Therefore, it was decided to combine all products in one database to calculate compliance.

For a considerable number of products, not all critical nutrient data were available. In particular, this was the case for data from India and China. As it is not possible to calculate compliance by including products that lack nutrient data, these products were excluded (known as the listwise deletion method). This might have introduced bias, as there is a chance of an unknown systematic common denominator of excluded products existing. The database of the Indian Nutrition Institute and the USDA database were used to fill this gap by comparing the compliances within these databases with those of The George Institute. The lack of Chinese data was compensated for by using the draft criteria for the Chinese Smart Choice organization that was developed in close collaboration with Choices International and is based on the same principles (such as the 10–20% rule and the product group specific cut-off values) [51].

### 5.2. Summary

These newly developed criteria will serve as an international standard and provide a guiding framework for food and nutrition policies, as well as front-of-pack labeling, to be implemented not only internationally but also at the country or national levels by organizations and food companies. As the composition of food products may differ from the international situation according to consumer taste, the international criteria can be adapted to meet country-specific needs. The international criteria were developed to contribute to the prevention and control of non-communicable diseases. The adaptation at the national level should take national food and nutrition priorities into consideration, particularly in low- and middle-income countries. These can also be adapted for the development of strategies to prevent other forms of malnutrition. Choices strongly recommends this adaptation to be implemented by a national scientific committee consisting of leading independent scientists. A protocol for such adaptation is available upon request.

## 6. Choices International Foundation

Choices International Foundation is a non-governmental organization that aims to help shape a food system that supports consumers in making healthy food choices by making the healthy choice the easy choice. It does so by facilitating partnerships between multiple stakeholders such as governments, scientists, NGOs and the industry in order to achieve a national program of nutrition improvement. The international criteria are one of Choices’ core competences that are deployed to function as a basis for nutrition policies such as front-of-pack nutrition labeling. Choices International Foundation is funded for a large part by FrieslandCampina, PepsiCo, and Unilever. These companies are not involved in the governance of the foundation.

## Figures and Tables

**Figure 1 nutrients-12-02774-f001:**
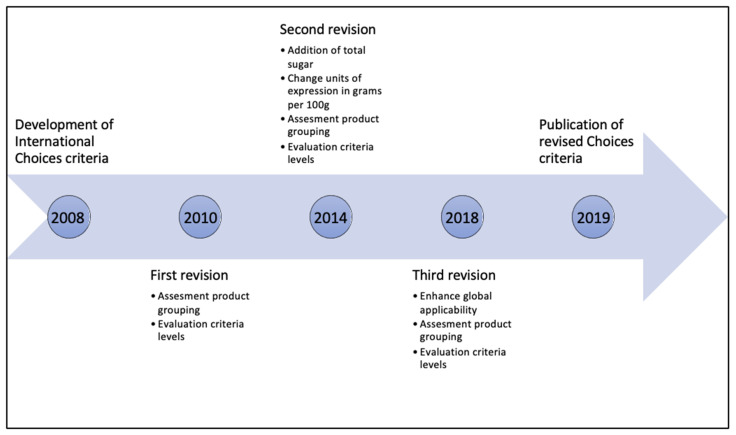
Timeline of the development and revisions of the International Choices criteria.

**Figure 2 nutrients-12-02774-f002:**
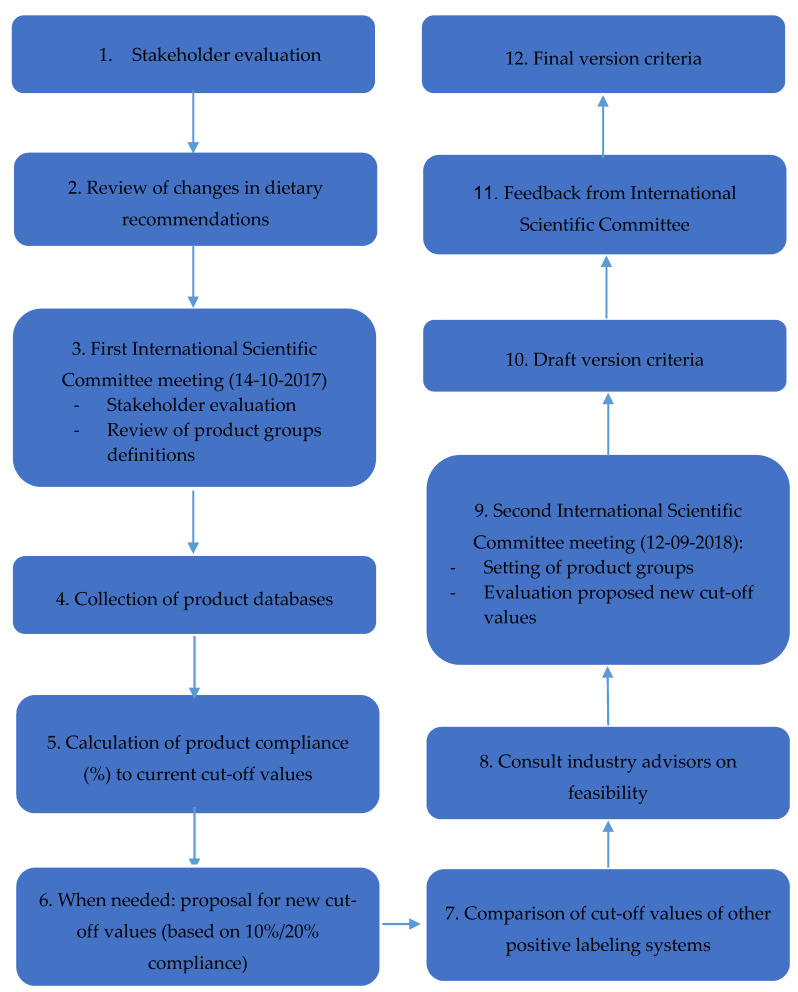
Flow chart of criteria revision process. September 2017–early 2019.

**Table 1 nutrients-12-02774-t001:** Summary of characteristics of GI databases.

	GI Database 1 (Obtained 2017)	GI Database 2 (Obtained 2018)
**Source**	The George Institute FoodSwitch database containing data from 22 largest food and beverage producers checked supplemented with data by these producers	The George Institute FoodSwitch data from global and local food producers
**Number of Choices product groups**	25	8
**Total number of products**	22,697	45,593
**Countries of origin**	Australia, China, Hong Kong, India, New Zealand, South Africa, UK, and USA	Australia, Hong Kong, India, South Africa, UK, and USA
**Nutrient data**	SAFA, Sodium, Total Sugar, Fiber	SAFA, Sodium, Total sugar

**Table 2 nutrients-12-02774-t002:** Critical nutrients.

	None	SAFA	TFA	Sodium	Sugar	Fiber	Energy
**BASIC PRODUCT GROUPS**							
**Fruits and Vegetables**							
Fresh or fresh frozen fruits, vegetables	x						
Processed and dried fruits and vegetables				x	x	x	
Processed beans and legumes		x		x		x	
**Water**							
Plain water, tea and coffee				x			
**Nuts and Seeds**							
Processed and unprocessed nuts and seeds		x		x			
**Sources of Carbohydrates**							
Plain tubers used as staple	x						
Processed tubers used as staple		x		x		x	
Plain noodles and pasta		x		x		x	
Flavored noodles and pasta		x		x		x	
Grains				x		x	
Bread		x		x	x	x	
Breakfast cereal products		x		x	x	x	
**Meat, Fish, Poultry, Eggs,**							
Unprocessed meat, poultry and eggs		x					
Processed meat and meat products		x		x			
Fresh, frozen, or processed seafoods		x		x			
Insects		x		x			
**Dairy**							
Milk (products)		x			x		
Cheese (products)		x		x			
**Oils, Fats, and Fat-Containing Spreads**							
Oils, fats and fat-containing spreads		x		x			
**Meals**							
Main meals		x		x	x		x
Sandwiches and rolls		x		x	x		x
Soups				x	x		
**NON-BASIC PRODUCT GROUPS**							
Meal sauces		x		x	x		
Emulsified sauces		x		x	x		x
Dark sauces				x	x		
Other sauces (water-based)				x	x		
Snacks		x	x	x	x		x
Fruit and vegetable juices					x		
Non-dairy milk substitutes		x			x		
Beverages					x		

SAFA: saturated fatty acid; TFA: trans fatty acid. Critical nutrients were specifically defined by expert opinions of the International Scientific Committee (ISC) members for each product group and form the most relevant and usually the most limiting factor for compliance.

**Table 3 nutrients-12-02774-t003:** Results of product group classification revision.

Category	Former Product Group (2016)	New Product Group (2019)
Sources of Carbohydrates	Potatoes (unprocessed)	Plain tubers used as staple
		Processed tubers used as staple
	Potatoes (processed), pasta, and noodles	Plain noodles and pasta
		Flavored noodles and pasta
	Rice	Grains
	Grains and cereal products—wheat-based	Bread
	Grains and cereal products—non-wheat products	Breakfast cereal products
	Bread	
	Breakfast cereal products	
Meat Substitutes, Such as Tempeh, Tofu, and Dairy-Based Meat Substitutes	Processed meat, meat products, and meat substitutes	Classified by source:
		tempeh, tofu, processed beans, and legumes
		Dairy-based meat substitutes: milk (products)
Seafood	Fresh or fresh frozen fish, shellfish, and crustaceans	Fresh, frozen or processed seafood
	Processed fish or fish products	
Edible Insects	Not classified	Insects
Milk and Milk Substitutes	Milk (products)	Milk (products)
		Non-dairy milk substitutes
Meals	Main course	Main meals
	Sandwiches/rolls	Sandwiches and rolls
	Mixed salads	
	Small meals	Soups
Water-Based Sauces	Other sauces (water based)	Dark sauces
		Other sauces (water-based)
Snacks	Snacks	Savory snacks
		Sweets snacks
Bread Toppings, Such as Fruit Spreads, Hummus, Tahini, Peanut Butter	Bread toppings	Fruit spreads: processed and dried fruits and vegetables
		Hummus: processed and dried beans and legumes
		Tahini, peanut butter: processed and unprocessed nuts and seeds

**Table 4 nutrients-12-02774-t004:** Choices International Criteria 2019.

	SAFA (g/100 g)	TFA (g/100 g)	Sodium (mg/100 g)	Added Sugars (g/100 g)	Total Sugars (g/100 g)	Fiber (g/100 g)	Energy (kcal/Portion or kcal/100 g)
**BASIC PRODUCT GROUPS**							
**Fruits and Vegetables**							
**Fresh or Fresh Frozen Fruits and Vegetables**	All products that do not contain additions comply						
**Processed and Dried Fruits and Vegetables**	≤1.1	≤0.1	≤100	Not added	≤10.0 (vegetables)	≥1.0	N/A
					or		
					≤17.0 (fruits)		
**Processed and Dried Beans and Legumes**	≤1.1	≤0.1	≤200	≤2.5	≤5.7	≥3.5	N/A
**Water**							
**Plain Water, Tea, and Coffee**	Not added	Not added	≤20	Not added	N/A	N/A	N/A
**Nuts and Seeds**							
**Processed and Unprocessed Nuts and Seeds**	≤8.0	≤0.1	≤100	Not added	≤7.5	N/A	N/A
**Sources of Carbohydrates**							
**Plain Tubers Used as Staple**	All products that do not contain additions comply						
**Processed Tubers Used as Staple**	≤1.1	≤0.1	≤100	Not added	≤3.0	≥2.7	N/A
					For sweet potatoes: ≤6.5		
**Plain Noodles and Pasta**	≤1.1	≤0.1	≤100	Not added	≤3.0	≥2.7	N/A
**Flavored Noodles and Pasta**	≤2.0	≤0.1	≤500	Not added	≤4.0	≥2.7	N/A
**Grains**	≤1.2	≤0.1	≤100	Not added	≤4.5	≥1.8	N/A
**Bread**	≤1.1	≤0.1	≤450	≤4.0	≤6.0	≥4.0	N/A
**Breakfast Cereal Products**	≤3.0	≤0.1	≤400	≤15	≤17	≥6.0	N/A
**Meat, Fish, Poultry, Eggs, and Meat Substitutes**							
**Unprocessed Meat, Poultry, and Eggs**	≤3.2	≤0.1	≤150	Not added	N/A	N/A	N/A
**Processed Meat and Meat Products**	≤5.0	≤0.1	≤450	≤2.5	≤2.5	N/A	N/A
**Fresh, Frozen, or Processed Seafood**	≤4.0	≤0.1	≤300	Not added	N/A	N/A	N/A
**Insects**	≤3.2	≤0.1	≤200	Not added	N/A	N/A	N/A
**Dairy**							
**Milk (Products)**	≤1.4	≤0.1	≤100	≤5.0	≤11.0	N/A	N/A
**Cheese (Products)**	≤14.0	≤0.1	≤750	Not added	N/A	N/A	N/A
**Oils, Fats and Fat-Containing Spreads**							
**Oils, Fats and Fat-Containing Spreads**	≤28.0	≤0.5	≤160	Not added	≤1.5	N/A	N/A
**Meals**							
**Main Meals**	≤2.0	≤0.15	≤240	≤3.0	≤5.0	≥1.2	≤600 kcal/portion
**Sandwiches and Rolls**	≤2.2	≤0.15	≤450	≤8.0	Sandwiches: ≤10.0	≥1.4	≤350 kcal/portion
					Rolls: ≤8.2		
**Soups**	≤1.1	≤0.1	≤250	≤1.5	≤4.0	N/A	≤100 kcal/100 g
**NON-BASIC PRODUCT GROUPS**							
**Meal Sauces**	≤1.1	≤0.1	≤400	≤2.5	≤6.0	N/A	N/A
**Emulsified Sauces**	≤3.0	≤0.3	≤700	≤10.0	≤10.0	N/A	≤350 kcal/100 g
**Dark Sauces**	N/A	N/A	≤3000	N/A	≤16	N/A	N/A
**Other Sauces (Water-Based)**	≤1.1	≤0.1	≤750	N/A	≤16	N/A	≤100 kcal/100 g
**Savory Snacks**	≤4.0	≤0.4	≤400	≤4.0	≤4.0	N/A	≤110 kcal/portion
**Sweet Snacks**	≤6.0	≤0.4	≤200	≤20	≤20	N/A	≤110 kcal/portion
**Fruit and Vegetable Juices**	≤1.1	≤0.1	≤100	Not added	≤12.0	≥0.75	≤48 kcal/100 g
**Non-Dairy Milk Substitutes**	≤1.1	≤0.1	≤100	N/A	≤5.0	N/A	≤40 kcal/100 g
**Beverages**	≤1.1	≤0.1	≤20	N/A	≤2.5	N/A	≤10 kcal/100 g
**All Other Products**	≤1.1 g/100 g or ≤10 en%	≤0.1 g/100 g or ≤1.0 en%	≤100 mg/100 g	≤2.5 g/100 g or ≤10 en%	N/A	N/A	N/A

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
