# Peer review of "Periodic Revisions of the International Choices Criteria: Process and Results"

_nutrients, 2020, doi:10.3390/nu12092774_

Round 1
Reviewer 1 Report
This is a well written and clear paper describing updates to the Choices Programme. The paper is clear and describes updates to the Programme, rationale and implications. My only comments are:
- please be more clear throughout regarding 'snacks' and what you mean by snacks - e.g. an apple can be a snack.
- for certain foods e.g. seafood, care is needed as they can contain constituents for which care is needed from a toxicological perspective and particularly for certain population groups.
Author Response
Dear reviewer 1,
Thank you so much for your compliments and for reviewing the paper. Apologies for the delay in my response. Because of health reasons it took longer for me to process the proposed feedback.
I made changes according to your suggestions in line 55 and 73.
Thanks again for taking the time to review. This is well appreciated!
On behalf of all co-authors,
Kind regards,
Sylvie van den Assum
Scientific secretary at Choices International Foundation
Reviewer 2 Report
The article is a largely well-written description of the changes in classification of nutrient value of food items by the organization Choices International Foundation from 2010 to the present. The decision process in changing the classification is very well described.
The manuscript lacks a better description of what type of organization Choices International Foundation actually is – an NGO?, a government-backed organization? a food industry-backed group? What has been the impact of the organization – has its criteria been used in any countries or regions of the world? Why is there no apparent presence in North America? Do the authors have any conflicts of interest?
Lines 279-283 is unclear, including the example of basic oatmeal. Were they concerned that fat content of oatmeal was too high to fit a healthy choice? Please describe in more detail.
Table 3: Shouldn’t the new product group for milk be both milk products AND non-dairy milk substitutes?
Author Response
Dear reviewer 2,
Thank you so much for your compliments and for reviewing the paper. Apologies for the delay in my response. Because of health reasons it took longer for me to process the proposed feedback.
I made changes according to your suggestions in table 3, and line 289 and 528.
Thanks again for taking the time to review. This is well appreciated!
On behalf of all co-authors,
Kind regards,
Sylvie van den Assum
Scientific secretary at Choices International Foundation
Reviewer 3 Report
- Nice discussion/inclusion of comparsion with other systems - more would help in table form; same with capturing a timeline history of the Choices program
Author Response
Dear reviewer 3,
Thank you so much for your compliments and for reviewing the paper. Apologies for the delay in my response. Because of health reasons it took longer for me to process the proposed feedback.
I made changes according to your suggestions and added a timeline depicting the development and revisions of the Choices International criteria. I chose not to add a table explaining the differences between other systems, as a comparison between systems was not a main focus of this paper and it would in my opinion result in a incomplete table. Similar comparisons have been performed elsewhere. (references 3, 40 and 41).
Thanks again for taking the time to review. This is well appreciated!
On behalf of all co-authors,
Kind regards,
Sylvie van den Assum
Scientific secretary at Choices International Foundation